# The Carbon Footprint of Marathon Runners: Training and Racing

**DOI:** 10.3390/ijerph18052769

**Published:** 2021-03-09

**Authors:** Laurent Castaignède, Frederic Veny, Johnathan Edwards, Véronique Billat

**Affiliations:** 1Laboratory of Analysis of the Environmental Impact of Activities, Products and Projects, BCO2Engineering, 33200 Bordeaux, France; 2BillaTraining SAS, 94140 Alfortville, France; 3Training and Research Unit in Basic and Applied Sciences University Evry-Paris-Saclay, 91000 Evry-Courcouronnes, France; 4Faculty of Motricity Sciences Teaching Unit in Physiology and Biomechanics of Movement, University of Bruxelles,1070 Bruxelles, Belgium

**Keywords:** carbon emissions, carbon impact, carbon calculator, environmental impact of sport

## Abstract

Marathon running leaves a significant carbon footprint regarding CO_2_ emissions; for example, 37 percent of New York Marathon participants travel internationally to New York. The aim of this study is to estimate the CO_2_ footprint of a person training and competing in a marathon; we will also propose methods to minimize the CO_2_ footprint because of transportation. In addition, we also examine the influence of food practices and hygiene on training and racing a marathon. Methods: We estimated the annual carbon footprint of one person taking part in a marathon. We considered all training, racing, and travelling (local and international) for one person (we are going to give him the first name of “Henri”), and then compared his CO_2_ footprint with his colleagues playing tennis and soccer. The excess CO_2_ footprint whilst running and for shoes, clothing, books, magazines, insurance, travel, hygiene, laundry, and resources for electronics and additional food consumed were calculated. For competitions, we estimated and compared the CO_2_ emission from transportation to national vs. international marathon (New York). Results: We estimated that our runner emitted 4.3 tons of CO_2_ equivalent (CO_2_e), including all greenhouse gases. A transatlantic flight to New York corresponded to 3.5 tons CO_2_, which is 83% of the annual carbon footprint of an average French citizen which is about 11 tons CO_2_e/year. This leads to a sudden 40% increase in Henri’s annual carbon footprint. Conclusions: By focusing on the additional carbon footprint from one year of marathon training and racing, and traveling locally versus internationally, this sport still has a potentially significant carbon footprint that runners and race organizers ought to consider. We wanted to answer a growing question of marathon runners who are wondering about the carbon footprint of their sports practice in following with a new environmentalist trend that considers not traveling anymore to participate in marathons and to stay local. However, the representativeness in the selection of calculation objectives is very low. There is no need for statistics since this study is a theoretical simulation of traditional training and competition practices of marathon runners.

## 1. Introduction

A carbon footprint is composed of the “total set of greenhouse gas emissions (CO_2_e or carbon dioxide equivalent) caused directly and indirectly by an individual, organization, event or product” [1]. Despite the health benefits of running, the people involved with sports often engage in practices that produce large amounts of CO_2_ emission [2,3]. The estimated quantity of CO_2_ emissions from sports practices such as transportation, the construction of sports facilities, and the production of sporting goods and services [4] are a significant threat to the quality of the natural environment [5,6]. Modes of transportation used by spectators and the athletes are believed to be the bulk of CO_2_ emissions [7,8].

The CO_2_ emissions implications of training and competing in a marathon have not been studied. Previous studies have highlighted that sporting events have become a negative contributor towards environmental degradation [2,9]. Indeed, the carbon footprint associated with sport participation has been a significant source of CO_2_ emissions, (i.e., 8% of overall emissions for a German adult). Interestingly, non-mainstream individual sports such as diving, golf and surfing leaves the three highest individual carbon footprints. The greenhouse gas emissions from more universal sports activities, such as cycling and walking, are not negligible (because of additional fueling, walking, and cycling). Replacing short car trips with cycling or walking does not significantly produce a carbon emissions savings.

Nowadays, sport is no longer considered to be separated from environment [10], and sports participation results in a disproportionate consumption of raw materials, traffic congestion, related air pollution, exhausting local water supplies, and a challenge around waste disposal [11,12]. However, increasingly there is a sociology of sports’ awareness with the environment [13]. The notion of “environmental waves” of sport has been proposed to understand the past, present, and future of environmental sustainability [14,15]. There is a need for sports to find a balance between the health benefits and the associated CO_2_ emissions. Modifications to equipment, infrastructure, materials, and industrial food are all current areas of interest [16,17]. Sustainable development of sports tourism as it relates to a positive carbon footprint is currently being investigated. 

In addition, the impact of new walking and cycling infrastructures (brides and paths) on CO_2_ emissions from motorized only showed minor effects on CO_2_ emissions; living near the infrastructures nor using it, predicted significant changes in CO_2_ emissions [18]. Including active mobility sports such as cycling has been reported to be an additional source of CO_2_ emissions due to fueling, walking and cycling. Human-powered locomotion is associated with non-negligible greenhouse gas emissions [19]. 

Marathons offer a unique opportunity to study the balance between sports participation and CO_2_ emissions. Marathon participation has exponentially increased over the last 2 decades and the CO_2_ emission associated with marathon events has exploded. The additional CO_2_ emissions associated with training and competing in a marathon should be potentially given that runners often train locally, consume less industrial nutrition, and use domestic transportation. 

Our aim of this study is to test the hypothesis of a person who is training and competing in marathons locally (within 1000 km) and the associated CO_2_ emissions related to this athletes’ activities and preparation for his marathon. We also compared the CO_2_ emissions of the athlete competing in a marathon locally versus travelling internationally to New York.

## 2. Materials and Methods

A carbon footprint is the total greenhouse gas (GHG) emissions or assessment caused by an individual, event, organization, service, or product which can be limited to its life cycle or one year. GHG are expressed as carbon dioxide equivalent (CO_2_e). The goal of the study is to focus on the primary sources of total CO_2_ emissions and how to reduce future impact by direct or indirect modifications. Several categories must be taken into consideration to evaluate GHG as it related to sports activity, such as raw energy consumption, raw materials and goods, food, services, transportation, travel, waste management, and equipment. To calculate the impact of the total emissions in equivalent CO_2_ mass, the data are multiplied by emission factors. Our calculations are in line with IPCC (Intergovernmental Panel on Climate Change) works and publications which provide regular assessments of the scientific basis of climate change, its impacts and future risks, and options for reducing GHG emissions on many levels.

The aim of this study is to determine the carbon footprint of a typical French marathon runner preparing for a local marathon over a one-year period. We evaluated the results and costs associated with the decision to compete locally versus internationally and assumed that our runner will live domestically and commute via conventional French transit systems. We then compared the results of our French marathon runner to his running colleagues as well to the other choices related to competing in a marathon. Finally, we will determine whether these observed activities are in line with reducing GHG emissions as it relates to the medium-term stabilization of climate change.

### 2.1. Marathon Runner Carbon Footprint

Our French marathon runner, Henri, lives in Alfortville, which is a Paris-suburb about 7 km east of Paris-Notre Dame, France. On January 1st, Henri decides to run a marathon by the end of the year, and he chooses the New York City Marathon. In preparation, Henri runs nearly every day, competes in several local races, including the Paris Marathon in April. His programs consider all purchases, nutrition changes as it relates to his marathon preparation, which are grouped into fifteen categories:Shoes: 4 pairs of running shoesClothes: tee shirts (5), running shorts (3), running socks (10), water-proof track suitMiscellaneous running supplies: Camel-Back^®^ hydration system, head lamp (Nao-Petzl France^®^), Garmin^®^ GPS-watch, 1 running book (The Science of the Marathon©), running magazine subscriptions (Wider outdoor^®^ and Runners World^®^)Daily runs: running 20 km to work (La Defense) each morning (instead of using the subway station)Weekend runs: 15 km near his homeSports infrastructure: negligible because he runs existing on roads or tracks and we assume that he does not damage themNutrition: besides his regular diet, he adds quinoa pasta (100 g), 4 eggs, meats (150 g), nuts (100 g), and prunes (250 g)Hygiene: 5 additional hot showers per weekLaundry: 2 additional loads of laundry each monthIT: 2 h of additional internet, computer, and smartphone use Running Race fees: running license, entry fees to three 10 km races, 3 half-marathons, 1 trail race, and the Paris Marathon (all races are local). He travels to these events by car or via public transportation. New York Marathon: race fees, lodging, food, economy class round trip airfare to New York (we assume a 50% chance his wife will accompany him; thus we include half of the impact of a second airline ticket).

Its CO_2_ respiration additional emission is neglected because it corresponds to crop absorption on fields, as a carbon cycle effect–would omit this because the CO_2_ emissions of produce transportation is not negligible and very difficult to account for.

The assessment will be presented with the six standard carbon footprint categories (Pandey et al., 2010):Energy: heat energy and electricity consumption.“Intrants”: goods, foods, and services purchases.Transportation: goods transportation.Travel: people traveling by mechanized systems.Waste: waste management.Immobilization: infrastructure amortization.

### 2.2. Changing the Carbon Footprint

We change Henri’s race location from the New York City Marathon to a closer city (1000 km round-trip) to estimate the decrease in the carbon footprint. 

### 2.3. Other Leisure Sportive Activities Comparison

Finally, we will compare the activity of marathon running to tennis and soccer, but in different ways.

Choosing tennis as an alternative activity (at a similar level) would correspond to the purchases of different goods (racket, tennis clothes, balls, and stringing), tennis club membership, weekly practice, tournament fees, and tickets and travel to the “Roland Garros Open de France” tennis tournament.

Regarding soccer, we assume that the person is a fan and not a player. The soccer fan supports a professional team which competes at national and international level. This activity will comprise purchasing one stadium season pass (in the Parc des Princes, supporting Paris-Saint-Germain soccer team), a subscription to a dedicated sports channel, purchasing a new wide-screen television set (assuming his current television is not broken), purchasing additional clothes (team scarf, team shirt with the name of his preferred player), eating additional foods while watching matches (peanuts, beers, and champagne), traveling occasionally to follow his team through France (by train or car-sharing), traveling by plane to see two European championship matches, and using more internet. We also replaced the professional events by a national tennis match and to a local fan zone attendance.

## 3. Results

The carbon footprint according each sport and traveling conditions is expressed in absolute tCO_2_e and relative value to the annual French carbon footprint is indicated in Table 1.

### 3.1. With the New York City Marathon Option

The first evaluation of GHG representative emissions shows a total mass of 4.3 tons of CO_2_e (equivalent CO_2_ including the effect of all GHG emitted), as represented in Figure 1.

In this case, it shows the greatest impact in the travel category, with the New York trip representing nearly all (83%) of the whole carbon footprint (3.56/4.3).

If we compare this global impact to the French annual carbon footprint of 11 tCO_2_e, we note that this running activity represents, on average, an additional 40% GHG impact. This is mainly due to the transatlantic travel for the New York City Marathon. 

### 3.2. With the Local Marathon Option 

By replacing the New York trip with a closer destination corresponding to a train travel (we supposed a 1000 km round trip), the total mass is lowered 6 times less to 0.7 tCO_2_e, (Figure 2). Interestingly, the travels impacts are almost neutral: running daily to work instead of using public transportation almost compensates for the additional travel to go to races.

Figure 2 demonstrates that the first item is now the “intrants” category, which is mainly composed of electronics, race fees, and additional food. The second item is energy, mainly composed of additional showers.

### 3.3. Comparison of the Carbon Footprint Impact of Marathon Running to Other Leisure Sport Activities (Tennis and Soccer Fan)

Figure 3 shows the comparison of the footprint impact of marathon running with other leisure sport activities.

As shown in the Figure 3, the main impacts are still travelling, which are mainly composed of plane trips (to New York for the runner as noted before, zero for the tennis player and twice into Europe for the soccer fan). If we cancel these plane trips, and replace them with a national running race and to a local fan-zone event, the modified diagram is shown in Figure 4.

In this comparison, the soccer fan has the highest carbon footprint, mainly impacted by electronic purchases and attending events. The tennis player has the lowest carbon footprint, even considering the amount of travel to tournaments, because, with our hypothesis, he specifically does not purchase electronics (unless he wants a new wide screen television to better watch matches as is the soccer fan’s case).

## 4. Discussion

This article is the first to demonstrate that marathon training and competition is a low carbon footprint activity if you do not travel to the competition. The CO_2_ emissions implications of training and competing in a marathon have not been studied. Previous studies have highlighted that sporting events have become a negative contributor towards environmental degradation [2,9]. Indeed, the carbon footprint associated with sport participation has been a significant source of CO_2_ emissions, (i.e., 8% of overall emissions for a German adult). Interestingly, non-mainstream individual sports such as diving, golf and surfing leaves the three highest individual carbon footprints. The greenhouse gas emissions from more universal sports activities, such as cycling and walking, are not negligible (because of additional fueling, walking, and cycling). Replacing short car trips with cycling or walking does not significantly produce a carbon emissions savings. 

It is a response to a major concern of runners regarding the environment, since a movement is taking place in favor of localized competition, which is largely possible in the context of a sport that offers a great possibility of a wide range of competition throughout the national territory. It was a question of determining how much of this low carbon footprint is due to training and competition. This study provides answers to practitioners for a sport that is in full growth and does not require any particular infrastructure except for the least polluted environment possible. Furthermore, marathon training is an activity that is often integrated into the domestic travel of people running to work, which makes it possible to consider it as an active soft mobility in the same way as cycling. 

Indeed, the main result of this study shows that by considering the 6 standard carbon footprint categories, marathon running can reach 40% of the average annual French carbon footprint if the runner competes in an international marathon. If he chooses to run locally, the carbon footprint decreases to only represent 7% of the total carbon footprint. This last case is comparable with the tennis player (6% of the total carbon footprint carbon) and much less than the soccer fan (20%). 

The primary advantage of marathon running is that it was included in the active travel, even if we did not calculate the consequence of the positive effects on health outcomes. Indeed, even short distance travel by walking and cycling (less than 3 km) [20], shows little evidence to the effectiveness of active transport interventions for reducing obesity [21]. Higher relative VO2max values are associated with greater life expectancy [22].

Independent of the level of training, running could be included in public health strategies to reduce greenhouse-gas emissions in urban-land transport [23,24]. However, prior studies have shown the benefits of walking and cycling, but it is unclear how environmental interventions that might attract walkers and cyclists can reduce CO_2_ emissions from transportation. To encourage higher running participation, digital personalized programs [25] and security for walking paths are included in the walkability index, [26] and may be helpful to promote active transportation [18].

Accordingly, with prior studies assessing the carbon footprint of travel patterns of the English Premier Soccer League clubs [3] and other fan-clubs [7,8,16], the main category increasing CO_2_ emissions is the travel for all sports (Marathon, Tennis and Soccer Fan-83, 49 and 65% respectively). Travel CO_2_ emissions, however, become negligible for the marathon running when the travel is limited to 1000 km by train (1%). For the last case, the Intrants (goods, food, and services purchases) become the main carbon footprint category. The additional food for marathon runners is not important; even if we balance food intake and energy expenditure which would require less food with an additional energy savings [17]. However, there are now food supply shortages [27].

The carbon footprint of the playing tennis agrees with prior studies [28] and active sport enthusiast [29,30]:Transportation: goods transportation.Travels: travel by mechanized systems.Waste: waste management.Immobilization: infrastructure amortization.

The main finding of this study shows that carbon footprint of marathon running, and other sports activities depends on the amount and distance traveled. Travel represents 83% (with a transatlantic flight) or less than 1% (with a 1000 km train round-trip) of the carbon footprint in our marathon scenario, which also shows that if we compare these last two figures to an ideal carbon footprint, with the goal of climate stabilization, we should target a goal of 2 tons CO_2_e. The overall impact of these extra-activities (corresponding to 0.6 to 1 t CO_2_e/year) is therefore too much to reach this goal [14,15]). Giving up plane travel simply does not reach such a goal. In order to be compliant in climate stabilization, it is necessary to give up electronics, not renewing licenses and memberships, avoid adding additional showers, and take part almost only in local activities. Therefore, beside the personal footprint carbon, society must also consider that the mega sporting events cause a considerable impact on the environment [11,13] as in sports industry [5].

## 5. Conclusions

Marathon running is an activity that corresponds to a significant carbon footprint. A typical marathon runner can decrease his carbon footprint by 80% by choosing to take part in a local marathon event and avoiding a transatlantic flight. The global impact of this activity is not climate friendly with over 40% of the average annual impact of the current French citizen. Perhaps even this is too much. Even if a person only changes the destination for a train accessible one, that person still only achieves one-third of the average annual climate impact if we maintain the goal of stabilizing the global carbon impact. We need to reduce many other impacts, especially electronics purchases, energy consumption, and the distance traveled.

Marathon event organizers must reconsider the dynamics surrounding attracting international participants if they genuinely want to lower the carbon footprint of their events.

Comparisons of other activities such as tennis and a soccer fan reveal that these are very carbon intensive; especially when they include travel, many extra purchases, etc. This implies that all sports activities must take into consideration the global perspective if they want to address the global climate issue.

## 6. Limitation of This Study

We wanted to answer a growing question of marathon runners who are wondering about the carbon footprint of their sports practice with a new environmentalist trend that is considering not traveling anymore to participate in marathons and stay local. It was a question of demonstrating the practice of the marathon on the condition of taking part in local competitions (in sufficient supply due to the explosion of offers), will induce a carbon footprint as low as that reported in the literature for the practice of tennis in the context of a territory already well equipped in terms of terrain like Germany [28]

However, the representativeness in the selection of calculation objectives is very low. There is no need for statistics since this study is a theoretical simulation of traditional training and competition practices of marathon runners [31]. Additionally, we applied the official calculation method according to the standards [32].

Indeed, this paper does not have statistics but is a prospective study based on the training, competition and consumption practices of marathon runners based on a sociological study conducted on French marathon runners. We will thus specify within the limits of the study that this study cannot be immediately generalized to marathon runners worldwide, except to demonstrate that a local vs. international competitive practice will contribute to strongly decrease the carbon footprint of marathon practice which had, these last years, seen a tourist development with a strong valence of globalization with travel agencies offering stays whose final objective was the participation in a marathon (New York, Paris, London, Tokyo, etc.).

## 7. Perspective

Future work on the possibility of running in the case of marathon training in an urban environment will now have to address the question of the influence of pollution on the practice of training and potentially set limits on the intensity of effort as a percentage of the maximum oxygen consumption not to be exceeded according to a given pollution index.

## Figures and Tables

**Figure 1 ijerph-18-02769-f001:**
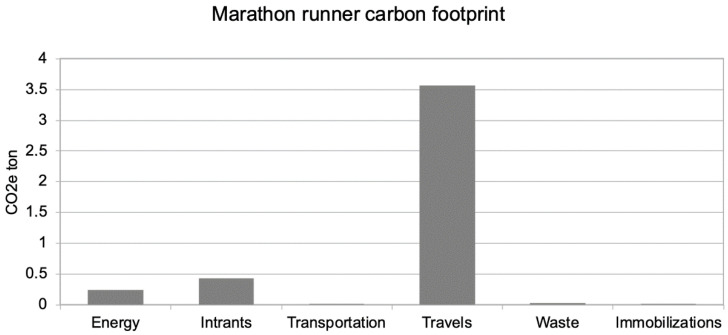
The carbon footprint of the marathon runner during 1 year of training and racing, including an international marathon in New York City.

**Figure 2 ijerph-18-02769-f002:**
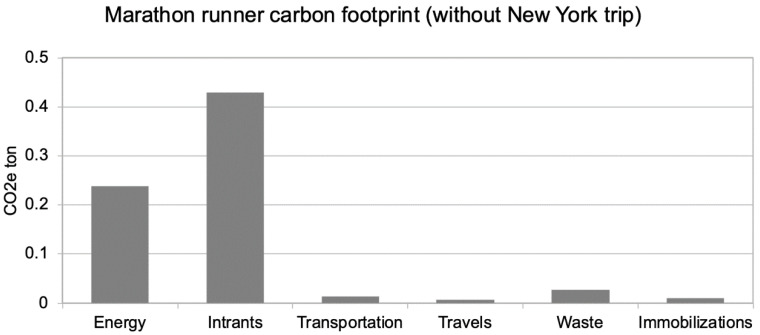
The carbon footprint of the marathon runner for one year of training and racing replacing the transatlantic marathon by a closer destination corresponding to a train travel (assuming a 1000 km round trip).

**Figure 3 ijerph-18-02769-f003:**
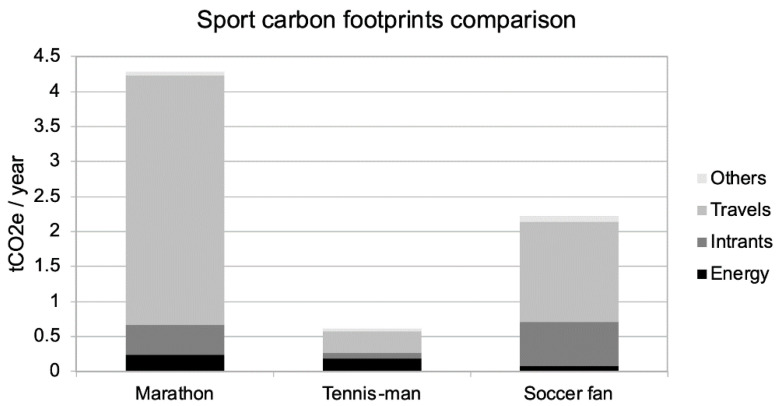
Comparison of the footprint impact of marathon running with other leisure sport activities and a transatlantic flight.

**Figure 4 ijerph-18-02769-f004:**
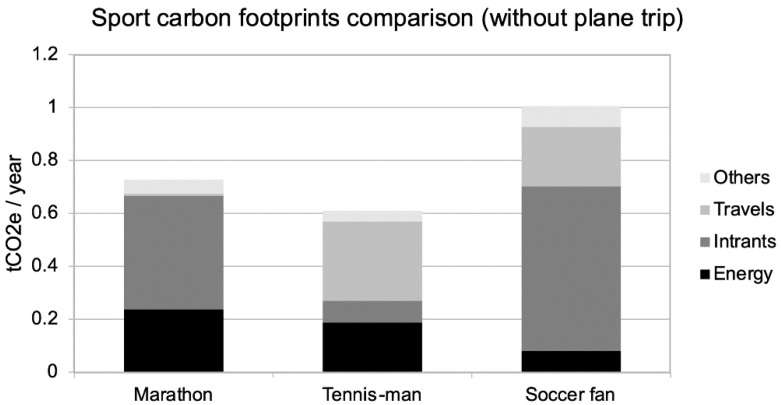
Comparison of the footprint impact of the practice of the marathon with other leisure sport activities without trans-Atlantic flight.

**Table 1 ijerph-18-02769-t001:** Synthesis of the carbon footprint according to the sport and travel conditions. In absolute value (tCO_2_e/year) and relative to the French carbon footprint.

	tCO_2_e/year							
Sports	Energy (tCO_2_/year % of the Total Activity)	Intrants	Transportation	Travels	Waste	Immobilization	Carbon FootPrint Total	% of Current French Carbon Footprint
marathon without transatlantic flight	0.24 33%	0.43 59%	0.01 2%	0.01 1%	0.03 4%	0.01 1%	0.73	7%
marathon with transatlantic flight	0.24 6%	0.43 10%	0.01 <1%	3.56 83%	0.03 1%	0.01 <1%	4.28	39%
Tennis	0.19 31%	0.08 13%	0.01 2%	0.30 49%	0.01 2%	0.02 3%	0.61	6%
Fan-club Soccer	0.08 4%	0.62 28%	0.01 <1%	1.44 65%	0.01 <1%	0.06 3%	2.22	20%
		tCO_2_e/year						

## Data Availability

The data presented in this study are available on request from the corresponding author.

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
