# Peer review of "The Carbon Footprint of Marathon Runners: Training and Racing"

_ijerph, 2021, doi:10.3390/ijerph18052769_

Round 1

Reviewer 1 Report

The major challenge is the small and limited case study of the present study. The case study highlights the extreme environmental impact from travel, but what exactly is added here? More discussion to highlight the study's novelty and contributions to the body of literature are needed before publication. 

Authors need to revisit the reference list. There are duplicates listed in the list that would require subsequent numbers to be changed in the manuscript. 

Author Response

Réponse R1

This article is the first to demonstrate that marathon training and competition is a low carbon footprint activity if you don't travel to the competition. It is a response to a major concern of runners regarding the environment, since a movement is taking place in favor of localized competition, which is largely possible in the context of a sport that offers a great possibility of a wide range of competition throughout the national territory. It was a question of determining how much of this low carbon footprint is due to training and competition. This study provides answers to practitioners for a sport that is in full growth and does not require any particular infrastructure except for the least polluted environment possible. Furthermore, marathon training is an activity that is often integrated into the domestic travel of people running to work, which makes it possible to consider it as an active soft mobility in the same way as cycling. Future work on the possibility of running in the case of marathon training in an urban environment will now have to address the question of the influence of pollution on the practice of training and potentially set limits on the intensity of effort as a percentage of the maximum oxygen consumption not to be exceeded according to a given pollution pollution index.

This article is the first to demonstrate that marathon training and competition is a low carbon footprint activity if you don't travel to the competition. The CO2e emissions implications of training and competing in a marathon have not been studied. Previous studies have highlighted that sporting events have become a negative contributor towards environmental degradation [9,2]. Indeed, the carbon footprint associated with sport participation has been a significant source of CO2 emissions, (i.e., 8% of overall emissions for a German adult). Interestingly, non-mainstream individual sports such as diving, golf and surfing leaves the three highest individual carbon footprints. The greenhouse gas emissions from more universal sports activities, such as cycling and walking, are not negligible (because of additional fueling, walking, and cycling). Replacing short car trips with cycling or walking does not significantly produce a carbon emissions savings.

It is a response to a major concern of runners regarding the environment, since a movement is taking place in favor of localized competition, which is largely possible in the context of a sport that offers a great possibility of a wide range of competition throughout the national territory. It was a question of determining how much of this low carbon footprint is due to training and competition. This study provides answers to practitioners for a sport that is in full growth and does not require any particular infrastructure except for the least polluted environment possible. Furthermore, marathon training is an activity that is often integrated into the domestic travel of people running to work, which makes it possible to consider it as an active soft mobility in the same way as cycling. Future work on the possibility of running in the case of marathon training in an urban environment will now have to address the question of the influence of pollution on the practice of training and potentially set limits on the intensity of effort as a percentage of the maximum oxygen consumption not to be exceeded according to a given pollution pollution index.

Reviewer 2 Report

The carbon footprint of the marathon is analyzed, and the research topic and conclusions of this paper are interesting. However, the representativeness in the selection of the calculation targets is very low. The sampling method from the population, statistical error, etc. should be considered in a research paper. Since these are not shown in this paper, the generality of the target of the assessment is unclear. Therefore, the evidence for the results of this study is insufficient.

Author Response

R2 The carbon footprint of the marathon is analyzed, and the research topic and conclusions of this paper are interesting : Thank you for these remarks, indeed we wanted to answer a growing question of marathon runners who are wondering about the carbon footprint of their sports practice with a new environmentalist trend that is considering not traveling anymore to participate in marathons and stay local. It was a question of demonstrating the practice of the marathon on the condition of taking part in local competitions (in sufficient supply due to the explosion of offers), will induce a carbon footprint as low as that reported in the literature for the practice of tennis in the context of a territory already well equipped in terms of terrain like Germany (28).

However, the representativeness in the selection of calculation objectives is very low. There is no need for statistics since this study is a theoretical simulation of traditional training and competition practices of marathon runners (31).

We applied the official calculation method according to the standards (32)

Wackernagel, M., Rees, W., 1996. Our Ecological Footprint: Reducing Human Impact on the Earth. New Society, Gabriola Island, BC.

 Population sampling method, statistical error, etc. should be considered in a research paper. As these elements are not included in this document, the generality of the evaluation target is unclear. Indeed, this paper does not have statistics but is a prospective study based on the training, competition and consumption practices of marathon runners based on a sociological study conducted on French marathon runners. We will thus specify within the limits of the study that this study cannot be immediately generalized to marathon runners worldwide, except to demonstrate that a local vs. international competitive practice will contribute to strongly decrease the carbon footprint of marathon practice which had, these last years, seen a tourist development with a strong valence of globalization with travel agencies offering stays whose final objective was the participation in a marathon (New York, Paris, London, Tokyo...).

Reviewer 3 Report

I enjoyed reading your manuscript, with interesting conclusions about the high carbon footprint derived from preparing and competing in a marathon (international or local), and its comparison with the carbon footprint of tennis and fan soccer. These findings can help athletes and sports fans reduce their carbon footprints.

The article is well written. The introduction provides sufficient background with updated references. The study was conducted using an appropriate methodology.

However, the abstract should be more concise, that excess of information can help to complete the introduction.

Please table 1 is duplicated, delete one.

It would be necessary to add the limitations of study.

Author Response

R3 I enjoyed reading your manuscript, with interesting conclusions about the high carbon footprint derived from preparing and competing in a marathon (international or local), and its comparison with the carbon footprint of tennis and fan soccer. These findings can help athletes and sports fans reduce their carbon footprints.

Many Thanks! indeed we wanted to answer a growing question of marathon runners who are wondering about the carbon footprint of their sports practice with a new environmentalist trend that is considering not traveling anymore to participate in marathons and stay local. It was a question of demonstrating the practice of the marathon on the condition of taking part in local competitions (in sufficient supply due to the explosion of offers), will induce a carbon footprint as low as that reported in the literature for the practice of tennis in the context of a territory already well equipped in terms of terrain like Germany (28).

The article is well written. The introduction provides sufficient background with updated references. The study was conducted using an appropriate methodology.

Many Thanks! Great

However, the abstract should be more concise, that excess of information can help to complete the introduction. OK we changed it according your remarks (in red in the revised version).

Please table 1 is duplicated, delete one. Sorry, it was done

It would be necessary to add the limitations of study. Of course we added the following chapter in the text: 6. Limitation of this study

We wanted to answer a growing question of marathon runners who are wondering about the carbon footprint of their sports practice with a new environmentalist trend that is considering not traveling anymore to participate in marathons and stay local. It was a question of demonstrating the practice of the marathon on the condition of taking part in local competitions (in sufficient supply due to the explosion of offers), will induce a carbon footprint as low as that reported in the literature for the practice of tennis in the context of a territory already well equipped in terms of terrain like Germany (28).

However, the representativeness in the selection of calculation objectives is very low. There is no need for statistics since this study is a theoretical simulation of traditional training and competition practices of marathon runners (31).

And we applied the official calculation method according to the standards (32)

Indeed, this paper does not have statistics but is a prospective study based on the training, competition and consumption practices of marathon runners based on a sociological study conducted on French marathon runners. We will thus specify within the limits of the study that this study cannot be immediately generalized to marathon runners worldwide, except to demonstrate that a local vs. international competitive practice will contribute to strongly decrease the carbon footprint of marathon practice which had, these last years, seen a tourist development with a strong valence of globalization with travel agencies offering stays whose final objective was the participation in a marathon (New York, Paris, London, Tokyo...).

Round 2

Reviewer 2 Report

It is important to clearly state the limitations of the study, and I agree with the authors' improvements. However, these limitations are not addressed in the abstract. The limitations should be clearly stated in the abstract. The title should also clearly state that it is a case study.

Author Response

It is important to clearly state the limitations of the study, and I agree with the authors' improvements. However, these limitations are not addressed in the abstract. The limitations should be clearly stated in the abstract. The title should also clearly state that it is a case study.

Of course, thank you for this suggestion 

we added this in the abstract.
